# Promoting Stochasticity for Expressive Policies via a Simple and Efficient Regularization Method

Qi Zhou    Yufei Kuang    Zherui Qiu    Houqiang Li    Jie Wang*
University of Science and Technology of China
{zhouqida, yfkuang, zrqiu}@mail.ustc.edu.cn
{lihq, jiewangx}@ustc.edu.cn

## Abstract

Many recent reinforcement learning (RL) methods learn stochastic policies with entropy regularization for exploration and robustness. However, in continuous action spaces, integrating entropy regularization with expressive policies is challenging and usually requires complex inference procedures. To tackle this problem, we propose a novel regularization method that is compatible with a broad range of expressive policy architectures. An appealing feature is that, the estimation of our regularization terms is simple and efficient even when the policy distributions are unknown. We show that our approach can effectively promote the exploration in continuous action spaces. Based on our regularization, we propose an off-policy actor-critic algorithm. Experiments demonstrate that the proposed algorithm outperforms state-of-the-art regularized RL methods in continuous control tasks.

## 1 Introduction

Reinforcement learning (RL) algorithms have achieved remarkable success, notably in video games [32, 31, 21] and robotic control [26, 33, 19]. However, traditional approaches to solving RL problems—that is, maximizing the cumulative rewards—often lead to deterministic policies [42], which are undesirable in many practical scenarios compared to their stochastic counterparts [54, 17, 33, 15, 10, 19, 20, 52, 7]. For example, stochastic policies are often better initializers for new tasks [17, 16] and are more robust in unexpected situations [17, 20, 52] than deterministic policies.

In order to learn stochastic policies, many methods draw on the entropy-regularized RL framework [34, 17, 19, 18, 25, 7, 36], which augments the standard RL objective (cumulative rewards) with an entropy term. Commonly used entropy terms for regularization include Shannon entropy [17, 19, 18, 51] and Tsallis entropy [25, 24, 7]. The former can improve both sample efficiency and stability [19, 20], while the latter can find a stochastic policy close to the optimal policy of the standard RL [52, 7]. However, in continuous control tasks, many entropy-regularized methods [20, 25] express policies by Gaussian distributions, which can hardly represent *complex* behaviors [17, 45, 28].

To tackle this problem, recent methods use expressive policy representations instead[17, 18, 51, 28]. Soft Q-learning [17] parameterizes the policies by a sampling network to capture multi-modal behaviors. Haarnoja et al. [18] proposed to use latent space policies to learn hierarchical policies. However, these methods often requires complex training procedures [45, 19]. Soft Q-learning [17] trains the sampling network by Stein variational gradient descent [27], which involves complicated variational inference [19, 45]. Training latent space policies [18] requires calculating the probability density with normalizing flows [38, 9], which leads to an expensive computational cost.

To address this challenge, we propose a novel **s**ample **b**ased **r**egularization (SBR) method, which is simple, efficient, and compatible with a broad range of expressive policy representations. The

---

major novelty of SBR is that, it promotes stochasticity by *encouraging sampled actions away from each other*, which does not require calculating the probability density. Based on SBR, we propose an off-policy algorithm, namely, **a**ctor-**c**ritic with generalized **e**nergy **d**istance (ACED). Experiments on six MuJoCo benchmarks show that ACED achieves stable performance improvement compared with the state-of-the-art regularized RL algorithm. Moreover, we evaluate ACED with several policy architectures [11, 38, 45] and empirically show that it can effectively improve the performance. We provide all proofs in the supplementary material.

## 2   Preliminaries

**Notation:** We consider an infinite-horizon discounted Markov decision process (MDP) with a continuous action space. An MDP is defined by a tuple $(\mathcal{S}, \mathcal{A}, P, r, \gamma)$, where $\mathcal{S} \subset \mathbb{R}^m$ is a continuous state space, $\mathcal{A} \subset \mathbb{R}^n$ is a compact action space that contains more than one action, $P : \mathcal{S} \times \mathcal{A} \times \mathcal{S} \to [0, \infty)$ represents the probability density function that is corresponding to the transition probability, $r : \mathcal{S} \times \mathcal{A} \to [R_{\min}, R_{\max}]$ is a reward function, $\gamma \in (0, 1)$ is a discount factor. Let $\pi : \mathcal{S} \to \Delta$ be a stationary policy, where $\Delta$ is a set of some Borel probability measures on $\mathcal{A}$. That is, $\pi(\cdot|s) \in \Delta$ measures the probability of actions at state $s$. For convenience, we overload the notation and let $\pi(\cdot|s)$ also denote the corresponding probability density function (PDF) without ambiguity. Let $\Pi$ denote the set of all possible policies. Equations and inequalities between functions are pointwise. For example, we say $v \le v'$ if $v(s) \le v'(s)$ for any $s \in \mathcal{S}$. Maximum and supremum of a function are also pointwise. For instance, $v^* = \sup_{\pi \in \Pi} v^\pi$ means $v^*(s) = \sup_{\pi \in \Pi} v^\pi(s)$ for any $s \in \mathcal{S}$. Given a (random) vector $x$, $[x]_i$ denotes its $i$th element. Given a distribution $q$ and an arbitrary random vector $X$ that obeys $q$, $[q]_i$ stands for the distribution of $[X]_i$.

**Regularized RL:** Regularized RL adds a regularization term to the standard RL objective (expected discounted return). That is, the regularized RL aims to solve the following problem

$$\pi^*_\alpha \triangleq \arg\max_{\pi \in \Pi} \mathbb{E}_\pi \left[ \sum_{t=0}^\infty \gamma^t \left( r(s_t, a_t) + \alpha \mathcal{F} \left( \pi(\cdot|s_t) \right) \right) \right],$$

where $\mathcal{F} : \Delta \to \mathbb{R}$ is a regularization term and $\alpha \ge 0$ is a hyperparameter. We can define the regularized state-action value function $Q$ and state value function $V$ by

$$Q^\pi_\alpha(s, a) \triangleq \mathbb{E}_\pi \left[ \sum_{t=0}^\infty \gamma^t \left( r(s_t, a_t) + \alpha\gamma \mathcal{F} \left( \pi(\cdot|s_{t+1}) \right) \right) \,\middle|\, s_0 = s, a_0 = a \right], \qquad (1)$$

$$V^\pi_\alpha(s) \triangleq \mathbb{E}_{a \sim \pi(\cdot|s)} \left[ Q^\pi_\alpha(s, a) \right] + \alpha \mathcal{F}(\pi(\cdot|s)). \qquad (2)$$

The optimal Q-value function is defeind by $Q^*_\alpha \triangleq \sup_{\pi \in \Pi} Q^\pi_\alpha$ and $V^*_\alpha \triangleq \sup_{\pi \in \Pi} V^\pi_\alpha$. When $\alpha = 0$, the regularized RL is essentially equivalent to the standard RL. In this condition, we omit the $\alpha$ in our notation. For example, $Q^*$ denotes the optimal state-action value function in standard RL.

## 3   From Entropy Regularization to Sample-Based Regularization

In this section, we focus on the regularization for control tasks, whose action spaces are continuous. First, we discuss a limitation of entropy regularization in Section 3.1. Then, we propose a sample-based regularization to alleviate the limitation in Section 3.2. Finally, We provide several instances of our regularization in Section 3.3, and then discuss their properties in Section 3.4 and Section 3.5.

### 3.1   Limitations of Entropy Regularization

Entropy-regularized RL often faces a dilemma between simple policy architectures, whose representational power is limited, and complex training procedures, whose costs are expensive.

Existing entropy regularizers take a general form of $\mathbb{E}_{a \sim \pi(\cdot|s)} \left[ f(\pi(a|s)) \right]$ [52]. For example, Shannon entropy is given by $\mathbb{E}_{a \sim \pi(\cdot|s)} \left[ \log \pi(a|s) \right]$. Most entropy-regularized algorithms [20, 25] estimate this kind of regularization terms by empirical average. This estimation method requires simple probability density functions for efficient calculation. Therefore, popular entropy-regularized methods [20, 25] often represent policies by simple distributions, such as Gaussians. However, recent work has

shown several disadvantages of simple policy representations, such as inefficient exploration [8], convergence to sub-optimal solutions [48] and lack of ability to capture multiple goals [17].

Some entropy-regularized methods use sophisticated policy architectures for strong representation power. However, they often require complex inferences to optimize policy [17] or inefficient calculations to estimate entropy terms [45]. For example, normalizing flow policies [51] requires a serial procedure to calculate the probability density, which brings a non-negligible extra cost.

Moreover, there are plenty of policy architectures that are hardly compatible with entropy-regularized RL, such as Dirac mixtures, whose (generalized) PDF is a convex combination of Dirac delta functions. Improving the compatibility of regularized RL with various policy architectures can promote future research on policy architectures and potentially improve the performance of regularized RL.

## 3.2 Sample-Based Regularization (SBR)

To alleviate the limitation described in Section 3.1, we propose **s**ample **b**ased **r**egularization (SBR), which does not require the PDFs to estimate regularization terms. SBR takes the following form:

$$\mathcal{F}(\pi(\cdot|s)) \triangleq \mathbb{E}_{a \sim \pi(\cdot|s)}\left[f(a)\right] + \mathbb{E}_{a,a' \sim \pi(\cdot|s)}\left[g(a,a')\right], \tag{3}$$

where both $f : \mathcal{A} \to \mathbb{R}$ and $g : \mathcal{A} \times \mathcal{A} \to \mathbb{R}$ are bounded continuous functions. We can estimate the regularization term simply and efficiently by an unbiased estimator

$$\hat{\mathcal{F}}(\pi(\cdot|s)) = \frac{1}{N}\sum_{i=1}^{N}\left(f(a^i) + \frac{1}{N-1}\sum_{j=1,j\neq i}^{N} g(a^i, a^j)\right), \tag{4}$$

where $N$ is the number of samples and $\{a^i\}_{i=1}^{N}$ are sampled from $\pi(\cdot|s)$.

According to the equation 3, our regularization consists of two parts. In the first part, the function $f$ defines an extra state-independent reward, which can represent the tendency to explore according to a priori knowledge. In the second part, the function $g$ models the mutual effect between performed actions. We can promote stochasticity by encouraging selected actions away from each other. Therefore, $g(a,a')$ should be positively correlated to the distance between $a$ and $a'$.

In order to make our regularizers analyzable in continuous action spaces, we make two assumptions.

**Assumption 1.** *The set $\Delta$ is weakly closed. That is, if $(q_i)_{i=1}^{\infty}$ weakly converges to $q$, then $q \in \Delta$.*

**Assumption 2.** *The function $Q_\alpha^\pi(s, \cdot)$ is L-Lipschitz for all $s \in \mathcal{S}$ and $\pi \in \Pi$.*

Assumption 1 usually holds in practice, e.g., when $\Delta$ is the set of all Dirac delta distributions. Assumption 2 is commonly-used in literature [37, 22] for the smoothness of the Q-value function.

Under these two assumptions, we show the optimality condition in our regularized RL framework.

**Theorem 1.** *Under Assumption 1 and 2, the optimal policy $\pi_\alpha^*$ must exist, and for any state it satisfies*

$$\pi_\alpha^*(\cdot|s) = \arg\max_{q \in \Delta} \mathbb{E}_{a \sim q}\left[Q_\alpha^*(s,a)\right] + \alpha\mathcal{F}(q). \tag{5}$$

## 3.3 Instances of Sample-Based Regularization

In this part, we first introduce **g**eneralized **e**nergy **d**istance (GED) and then propose two instances of SBR by using GED. GED is commonly used to test the equality of two distributions [39, 44]. Given two probability measures $q, p \in \Delta$, their generalized energy distance $d_\phi$ is defined by

$$d_\phi^2(q,p) \triangleq 2\mathbb{E}_{x\sim q, y\sim p}\left[\phi(\|x-y\|_2^2)\right] - \mathbb{E}_{x,x'\sim q}\left[\phi(\|x-x'\|_2^2)\right] - \mathbb{E}_{y,y'\sim p}\left[\phi(\|y-y'\|_2^2)\right] \tag{6}$$

where the function $\phi : \mathbb{R} \to \mathbb{R}$ is nonnegative and its derivative $\phi'$ is completely monotone [3]. That is, $\phi(z) \geq 0$ and $(-1)^n\phi^{(n+1)}(z) \geq 0$ hold for all $z > 0$ and nonnegative integers $n$. Previous work [3] has shown that $d_\phi$ is a distance metric on $\Delta$ and provided some instances of function $\phi$ (see Table 1). Based on GED, we can further define $\tilde{d}_\phi^2$ by $\tilde{d}_\phi^2(q,p) \triangleq \sum_{i=1}^{n} d_\phi^2([q]_i, [p]_i)$. We can show that $\tilde{d}_\phi$ is also a distance metric on a specific space $\Delta$ (see Theorem 2).

**Theorem 2.** *Assume that any distribution $q \in \Delta$ is a product of its margin distributions. Then $\tilde{d}_\phi$ is a metric on the set of probability measures $\Delta$.*

Table 1: Several examples of function $\phi$.

| Function $\phi$ | Condition | Function $\phi$ | Condition |
|---|---|---|---|
| $\phi(z) = z^\kappa$ | $0 < \kappa < 1$ | $\phi(z) = \log(1 + \kappa z)$ | $0 < \kappa$ |
| $\phi(z) = \frac{z}{z+\kappa}$ | $0 < \kappa$ | $\phi(z) = 1 - e^{-\kappa z}$ | $0 < \kappa$ |

Table 2: Instances of sample-based regularization. We *omit the constant term* in $\mathcal{F}(q)$. For example, we omit the constant term $\mathbb{E}_{a,a'\sim u}[\phi(\|a - a'\|_2^2)]$ when using $\mathcal{F}(q) = -d_\phi^2(q, u)$.

| $\mathcal{F}(q)$ | $f(a)$ | $g(a, a')$ |
|---|---|---|
| $-d_\phi^2(q, u)$ | $-2\mathbb{E}_{a'\sim u}\left[\phi(\|a - a'\|_2^2)\right]$ | $\phi(\|a - a'\|_2^2)$ |
| $-\tilde{d}_\phi^2(q, u)$ | $-2\sum_{i=1}^n \mathbb{E}_{a'\sim u}\left[\phi\left(([a]_i - [a']_i)^2\right)\right]$ | $\sum_{i=1}^n \phi\left(([a]_i - [a']_i)^2\right)$ |

Note that the Shannon entropy of a distribution $q$ can be regarded as the KL divergence of the uniform distribution $u$ from the distribution $q$ (added by an constant). Inspired by this fact, we define two regularization terms via $d^2(q, u)$, where $d$ can be $d_\phi$ or $\tilde{d}_\phi$. Then, the regularization term takes the form of SBR (see Table 2). In practice, the first regularization $d_\phi^2$ requires sampling from a uniform distribution to approximate $f$, which leads to a large estimation variance. In the contrast, we can solve the closed-form expression of the function $f$ when using $\tilde{d}_\phi^2$, which helps to reduce variance.

### 3.4 Theoretical Results

We provide our theoretical results for $d_\phi$ in this subsection while similar results for $\tilde{d}_\phi$ can be found in Appendix A. First, we show that we can control the stochasticity of the optimal policies by tuning $\alpha$ and provide a sufficient condition that ensures the optimal policies to be stochastic.

**Theorem 3.** *Assume that the optimal policy $\pi_\alpha^*$ exists for all $\alpha \geq 0$ and the uniform distribution belongs to the set of probability measures $\Delta$. If we use the regularizer $\mathcal{F}(q) = -d_\phi^2(q, u)$, then we have $\pi_\alpha^*(\cdot|s) \xrightarrow{d_\phi} u$ for all states when $\alpha \to \infty$.*

**Theorem 4.** *Suppose the action space $\mathcal{A}$ is convex and the set $\Delta$ contains all Borel probability measures on $\mathcal{A}$. Under Assumption 2, if we use the regularizer $\mathcal{F}(q) = -d_\phi^2(q, u)$ and choose function $\phi$ such that $\lim_{z\to 0^+} 2\alpha\sqrt{z}\phi'(z) > L$, then $\pi_\alpha^*(\cdot|s)$ will never be a Dirac delta distribution.*

Theorem 3 states that GED-based regularizers can bring active exploration when $\alpha$ is large enough. We illustrate Theorem 3 in Figure 1. Theorem 4 shows that we can ensure the stochasticity of the optimal policies by choosing proper function $\phi$, e.g., $\phi(z) = z^\kappa$ with $0 < \kappa < 0.5$.

Finally, we compare the performance of $\pi_\alpha^*$ and $\pi^*$ in the original MDP.

**Theorem 5.** *Assume that the optimal policy $\pi_\alpha^*$ exists and the action space satisfies $\mathbf{diam}(\mathcal{A}) \leq U$. If we use the regularizer $\mathcal{F}(q) = -d_\phi^2(q, u)$, then we have*

$$V^* - \frac{2\alpha}{1 - \gamma}\left(\phi(U^2) - \phi(0)\right) \leq V^{\pi_\alpha^*} \leq V^*. \tag{7}$$

Theorem 5 proposes a bound on the performance gap between the policy $\pi_\alpha^*$ and the policy $\pi^*$. It shows that the gap depends on $\Delta = \phi(U^2) - \phi(0)$. Therefore, though larger $\Delta$ can encourage actions away from each other better, it may lead to worse performance of $\pi_\alpha^*$.

### 3.5 Discussion on Regularization Terms

We discuss the properties of SBR intuitively and provide some empirical results in continuous-armed bandit tasks. In this part, we consider an additional regularizer that is based on Gini mean difference (GMD) [53]. This regularizer is given by $\mathcal{F}(q) \triangleq \mathbb{E}_{a,a'\sim q}[|a - a'|]$, and thus it is essentially the second part of GED-based regularizer when $\phi(z) = z^{0.5}$. Our discussion includes three parts.

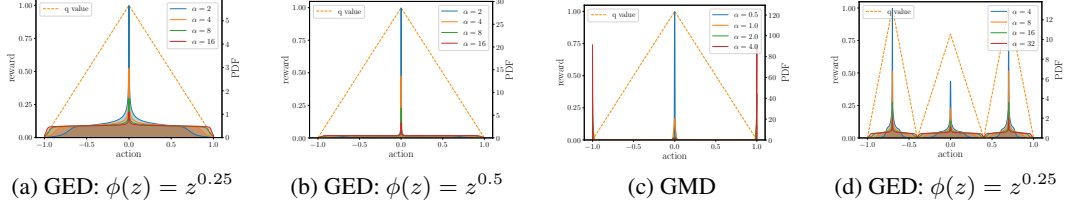

| (a) GED: $\phi(z) = z^{0.25}$ | (b) GED: $\phi(z) = z^{0.5}$ | (c) GMD | (d) GED: $\phi(z) = z^{0.25}$ |

Figure 1: Learned policy distributions in continuous-armed bandit tasks. The action space is a closed interval $[-1, 1]$. Figures (a), (b) and (d) correspond to GED-based regularization and figure (c) to regularization that is based on Gini mean difference (GMD) [53]. The dashed lines represent reward functions, and the solid lines denote the PDFs of learned policy distributions. When using GED-based regularization, the exploration becomes more active as $\alpha$ increases. However, when using GMD-based regularization, the performed actions gather near the boundary. Moreover, The subfigure (d) shows that GED-based regularization helps to capture multiple modes of near-optimal behaviors.

**Promoting stochasticity** We compare different instances of SBR under different $\alpha$ (see Figure 1). The GED-based regularization can effectively promote the stochasticity of learned policies while the GMD-based regularization does not perform well. Indeed, the GMD-based regularizer is the second part of the GED-based regularizer when $\phi(z) = z^{0.5}$. These results demonstrate that the first part of our regularizer (3) is significant for exploration. Compared with Figure 1a and 1b, using $\phi = z^{0.25}$ performs better than $\phi(z) = z^{0.5}$ in avoiding central tendency. A potential reason is that the value of $\phi(z) = z^{0.25}$ increases faster than the value of $\phi(z) = z^{0.5}$ as $z$ increases near the point $z = 0$.

**Learning multimodal behaviors** Consider a bandit task that takes optimal actions at $\pm 0.7$ and suboptimal action at $0$. We illustrate the optimal policy with GED-based regularizer in Figure 1d. The result shows GED-based regularizer helps to capture multiple near-optimal behaviors. We further provide an experiment in a 2D multi-goal environment to show this property (see Appendix C).

**Incorporating geometric information** Entropy regularizers mainly take account of the probability density but ignore the geometric information—how close two performed actions might be—to some extent [4]. In contrast, GED-based regularizers attach importance to the geometric information since their second parts encourage large distances between performed actions. We note that the geometric information is important in exploration. For example, considering the action space $\mathcal{A} = [0, 10]$, the entropy of the uniform distribution over $[0, 1]$ equals to that of the uniform distribution over $\bigcup_{i=1}^{10} [i - 0.1, i]$, while in practice we prefer to choose the latter as the exploration policy.

## 4   Algorithm

In this section, we propose the **a**ctor **c**ritic with generalized **e**nergy **d**istance (ACED), an off-policy RL algorithm that uses a GED-based regularizer. We show the pseudo code in Algorithm 1. We use neural networks to represent the regularized Q-function $Q_\psi$ and the policy $\pi_\theta$, where $\psi$ and $\theta$ are the parameters. We update these parameters by Adam [23]. We also use the target Q-value network $Q_{\bar\psi}$ whose parameters $\bar\psi$ is updated in a moving average fashion [26]. As with SAC [19], we use the clipped double Q-learning [12] and the automatic tuning for the hyperparameter $\alpha$ [20].

---

**Algorithm 1** ACED

**Input:** $\psi_1, \psi_2, \theta$

1: $\bar\psi_1 \leftarrow \psi_1, \bar\psi_2 \leftarrow \psi_2, \mathcal{D} \leftarrow \emptyset$
2: **for** each iteration **do**
3:     **for** each environment step **do**
4:         $a_t \sim \pi(\cdot|s_t)$
5:         $s_{t+1} \sim P(\cdot|s_t, a_t)$
6:         $\mathcal{D} \leftarrow \mathcal{D} \cup \{s_t, a_t, r(s_t, a_t), s_{t+1}\}$
7:     **end for**
8:     **for** each training step **do**
9:         $\psi_i \leftarrow \psi_i - \beta_Q \nabla J_Q(\psi_i)$ for $i = 1, 2$
10:        $\theta \leftarrow \theta - \beta_\pi \nabla J_\pi(\theta)$
11:        $\alpha \leftarrow \alpha - \beta_\alpha \nabla J_\alpha(\alpha)$
12:        $\bar\psi_i \leftarrow \tau \psi_i + (1 - \tau) \bar\psi_i$ for $i = 1, 2$
13:     **end for**
14: **end for**

**Output:** $\psi_1, \psi_2, \theta$

---

In this work, we use the regularizer $-\tilde{d}_\phi^2(q, u)$. The reason is that we can calculate the explicit expression of the function $f$ by integral, while using the regularizer $-d_\phi^2(q, u)$ may require an approximation of the function $f$. We detail our candidates for function $\phi$ and policy representation in Appendix B. In Section 5.1, we show that using $\phi(z) = z^{0.25}$ with a simple Dirac mixture policy can achieve state-of-the-art performance in the continuous control benchmarks.

Given $Q_{\bar{\psi}}$, $\pi_\theta$ and $\alpha$, we train $Q_\psi$ by minimizing the regularized Bellman residual

$$J_Q(\psi) = \mathbb{E}_\mathcal{D} \left[ \frac{1}{2} \left( r(s_t, a_t) + \gamma V_{\bar{\psi}}(s_{t+1}) - Q_\psi(s_t) \right)^2 \right], \tag{8}$$

where $\mathcal{D}$ denotes the replay pool, the expectation $\mathbb{E}_\mathcal{D}$ is actually the empirical average and the value function $V_{\bar{\psi}}$ is implicitly parameterized through the target Q-function parameters via Equation (2).

We apply the reparameterization trick to train $\pi_\theta$. Given $Q_\theta$ and $\alpha$, we maximize the objective

$$J_\pi(\theta) = \mathbb{E}_\mathcal{D} \left[ \mathbb{E}_{a \sim \pi_\theta(\cdot|s)} \left[ Q_\psi(s_t, a_t) \right] + \alpha \mathcal{F}(\pi_\theta(\cdot|s_t)) \right]. \tag{9}$$

We apply a gradient-based method to automatically tune the hyperparameter $\alpha$ similarly to SAC [20]. That is, given the current policy $\pi_\theta$, we tune $\alpha$ by minimizing the objective

$$J_\alpha(\alpha) = \alpha \mathbb{E}_\mathcal{D} \left[ \mathcal{F}(\pi(\cdot|s_t)) - T \right], \tag{10}$$

where $T$ is the target value of the regularization term. That is, we adjust the hyperparameter $\alpha$ to make $\mathbb{E}_\mathcal{D} \left[ \mathcal{F}(\pi(\cdot|s_t)) \right]$ approximately equal to the target value $T$ in expectation.

We use a minibatch to calculate the estiamted gradient $\hat{\nabla} J_Q(\psi)$, $\hat{\nabla} J_\pi(\theta)$ and $\hat{\nabla} J_\alpha(\alpha)$, and use the Equation (4) to estimate the regularization term and its gradient. We note that the extra time cost of multiple sampling is negligible (shown in Table 4) since this procedure can be easily parallelized.

There are three differences between ACED and SAC: 1) ACED is compatible with a broader range of policy architectures compared with SAC; 2) ACED uses GED-based regularizers while SAC uses entropy regularizers; 3) ACED does not need to calculate probability density while SAC needs to.

## 5  Experiments

Our experiments have three goals: 1) to test whether ACED can achieve state-of-the-art performance; 2) to show the computational efficiency of ACED; 3) to analyze the effect of each component in ACED. More experiments can be found in Appendix C.

In this section, we consider three GED-based regularizers, including **power-0.25** ($\phi(z) = z^{0.25}$), **power-0.5** ($\phi(z) = z^{0.5}$) and **log** ($\phi(z) = \log z$). We take into account four policy architectures, including squashed Gaussian (SG) [19], Dirac mixtures (DM), noisy networks (NN) [11] and generative models (GM) [14, 30]. For completeness, we provide details of experiments, including the implementation of different policy architectures, in the supplementary material.

### 5.1  Comparative Evaluation

We compare our algorithms with state of the art on six continuous control tasks in the MuJoCo control suite [49]. Our baselines include soft actor critic algorithm (SAC) [20], the state-of-the-art regularized RL methods; twin delayed deep deterministic policy gradient algorithm (TD3) [12], the state-of-the-art method that learns a deterministic policy; and deep deterministic policy gradient (DDPG) [26], an efficient off-policy algorithm. We stress that we did not tune the hyperparameters.

We report the results in Table 3 and Figure 2. All results are reported over six random seeds. Table 3 shows that *all four variants of ACED outperform SAC* in terms of averaged relative performance. Furthermore, we find that ACED outperforms SAC even when using the same policy architectures. We discuss the potential reasons in Appendix C. Figure 2 shows that ACED achieves stable improvement in sample efficiency compared to SAC. Moreover, ACED achieves the best final performance compared with all baselines except on the Swimmer-v2 task.

### 5.2  Computational Efficiency

We compare computational efficiency between ACED and SAC (see Table 4). We evaluate these two algorithms with different policy representations, including squashed Gaussian [20] and normalizing flow (NF) [28]. To avoid the impact of implementation on computing efficiency, we implement SAC using the same code framework as ACED. Results show that ACED can achieve higher computational efficiency than SAC, especially when using a complex policy architecture (NF). The main reason is that ACED does not need to compute the probability density. Moreover, using $N = 32$ requires only a little extra time compared with using $N = 2$ because the sampling is highly parallel.

Table 3: Relative performance of ACED compared to SAC. We report the best performance in 2000k-step training. The percentage corresponds to the ratio of the best score of ACED to that of SAC. Each variant is named with the rule "policy architecture"+"function $\phi$". For each task, we bold the best performance and use the corresponding variant to plot the ACED curve in Figure 2. This table shows the stable performance improvement of ACED compared to SAC.

| Variant | HCheetah | Ant | Walker2d | Swimmer | Hopper | Humanoid | Average |
|---|---|---|---|---|---|---|---|
| DM+log | 119.8% | 128.6% | **109.6**% | 76.3% | 96.3% | 117.5% | 108.0% |
| SG+log | **121.2**% | 125.4% | 102.8% | 91.9% | 97.8% | 123.1% | 110.4% |
| DM+power-0.25 | 116.7% | 122.2% | 94.9% | 97.5% | **113.4**% | 118.7% | 110.6% |
| SG+power-0.25 | 108.5% | **135.2**% | 101.5% | **100.1**% | 103.6% | **123.3**% | 112.0% |
| Average | 116.6% | 127.8% | 102.2% | 91.4% | 102.8% | 120.6% | 110.2% |

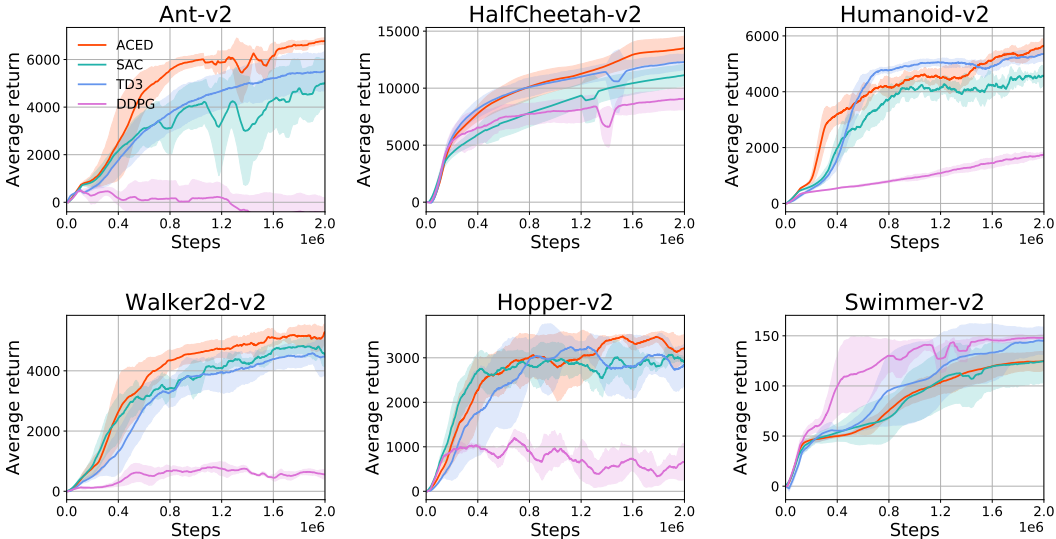

Figure 2: ACED versus SAC, TD3, DDPG on 6 Mujoco environments. The solid curves correspond to the mean and the shaded region to the standard deviation over 6 random seeds. We smooth curves uniformly for visual clarity. ACED (red curve) outperforms our baselines in most of benchmark tasks.

## 5.3 Ablation Study

We first analyze the sensitivity of ACED to the hyperparamter $N$. Then, we evaluate ACED with different regularizers and policy architectures. We report results for HalfCheetah-v2 in this subsection below and provide other results in Appendix C. We run each experiment with at least three seeds.

**Sample number** ACED uses multiple samples to estimate the regularization term. In Figure 3a, we show the performance of ACED with different $N$. Results demonstrate that ACED is insensitive to the hyperparameter $N$. Therefore, we recommend to use $N = 2$ for computational efficiency.

**Policy architecture** ACED is compatible with a broad range of policy architectures. We evaluate ACED with four architectures (see details in Appendix B). Also, we provide performance without regularization. Figure 3b shows that ACED can effectively improve the performance for all policy architectures. One of the main reasons is that GED-based regularization can promote the stochasticity and then bring active exploration. Therefore, GED-based regularizers are promising alternatives to entropy regularization for promoting stochasticity.

**Regularization term** We evaluate ACED with four different regularizers. In this experiment, *internal energy* [43] corresponds to the second part of the regularization term **power-0.25**. Figure 3c shows that using regularizers **log** and **power-0.25** achieve similar final performance while *internal energy* and **power-0.5** achieve relatively poor performance. These results demonstrate that the first part of SBR is also significant for exploration. Moreover, the results demonstrate that choosing function $\phi$ according to Theorem 4 can potentially improve exploration, since both $\phi(z) = \log z$ and $\phi(z) = z^{0.25}$ satisfy that $\lim_{x \to 0^+} \sqrt{x}\phi'(x) = \infty$ while $\phi(z) = z^{0.5}$ does not satisfy it.

Table 4: Comparison of computational efficiency between ACED and SAC in HalfCHeetah-v2 task. We run each experiments with one GPU (Nvidia Geforce RTX 2080Ti) and report the wall-clock time for 2000k-step training. Performance comparison for NF policies can be found in Appendix C.

| Algorithm | ACED | | | SAC | |
|---|---|---|---|---|---|
| Policy | SG ($N = 2$) | SG ($N = 32$) | NF ($N = 2$) | SG | NF |
| Time (h) | $9.80 \pm 0.04$ | $11.14 \pm 1.25$ | $12.27 \pm 0.80$ | $10.05 \pm 0.05$ | $18.75 \pm 0.03$ |

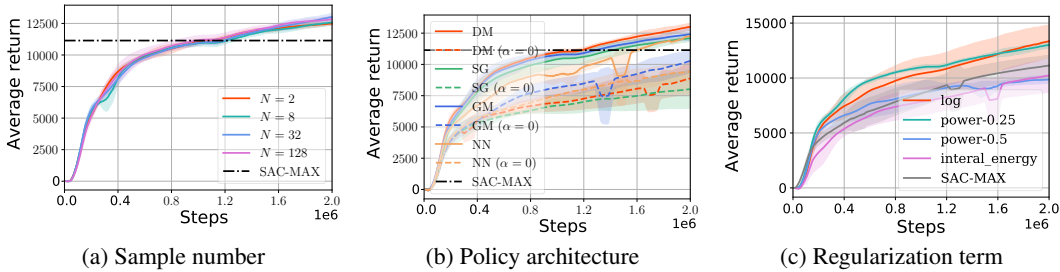

(a) Sample number     (b) Policy architecture     (c) Regularization term

Figure 3: Learning curves in HalfCheetah-v2 task. (a) ACED is insensitive to the hyperparameter $N$. We recommend to use $N = 2$ to reduce computational cost. (b) We evaluate four variants of ACED with different policy architectures. All the four variants outperform SAC. Moreover, the performance improvement is significant compared to algorithms without regularization ($\alpha = 0$). (c) ACED is sensitive to the choice of regularizers. We have provided suggestions on the choice in Section 3.

# 6 Related Work

In this section, we discuss related work, including regularization in RL and policy representations in continuous action spaces.

**Regularization in RL** Many RL methods use entropy regularization to learn stochastic policies [41, 40, 33, 34, 19, 17]. Recent work introduces entropy regularization into approximate policy iteration approaches [19, 25, 52] and shows the improvement in sample efficient and stability [19, 20]. Neu et al. [35] provide a unified view that considers entropy-regularized MDPs as a convex optimization problem. Ahmed et al. [1] analyze the impact of entropy regularization and empirically show that the entropy regularization smooth the optimization landscape. Yang et al. [52] propose a entropy-like regularzer form (shown in Section 3.1) and provide suggestions on the choice of regularizers. Geist et al. [13] provide generic theoretical results for regularized policy iteration.

**Policy Representations in Continuous Action Spaces** Previous work often present policies by simple parametric distributions, such as Gaussian [40, 41], Beta [6] and Delta [26, 12]. Representing policies by mixture distributions can capture moltimodal behaviors. The application of Gaussian mixture to SAC is widely studied [19, 18]. Many methods use Dirac mixtures by discretizing the action spaces [47, 2, 29]. Implicit distributions are commonly used in GAN [14, 30] and have been introduced into policy representation recently [45, 48]. Moreover, recent work use normalizing flow policies in both on-policy [46] and off-policy RL algorithms [51, 28, 18].

# 7 Conclusion

In this paper, we present a simple and efficient regularization method called Sample-Based Regularization (SBR). An appealing feature of SBR is that, we can estimate the regularization terms without knowing PDFs of policy distributions. Moreover, we propose several instances of SBR based on generalized energy distance and analyze their properties in continuous action spaces. Based on SBR, we present the actor critic with generalized energy distance (ACED), an off-policy regularized reinforcement learning algorithm. Our experiments show that ACED is comparable and often surpasses state of the art in six MuJoco benchmarking tasks. Our experiments demonstrate that SBR can improve the performance for a broad range of policy architectures. Theoretical analyses on the effect of policy architectures on performance are exciting avenues for future work.

## Acknowledgments and Disclosure of Funding

We would like to thank all the reviewers for their insightful comments. This work is supported by The National Key Research and Development Program of China (2017YFA0700800) and National Natural Science Foundation of China (61822604, 61836006, 61836011, U19B2026).

## Broader Impact

This work focuses on promoting stochasticity for expressive policies via sample-based regularization. It has the following potential positive impact: a). it encourages future research on more expressive policy architectures; b). it proposes a novel sample-based regularization method, which inspires future work to find more efficient regularizers in regularized RL algorithm; c). it promotes the application of RL algorithms in more complex control tasks. However, any reinforcement learning method runs the risk of being applied to military activities. Our work is no exception.

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
