[Supplementary Material]

# Promoting Stochasticity for Expressive Policies via a Simple and Efficient Regularization Method (Supplemental Material)

## A    Proof for Section 3

### A.1    Proof for Theorems in Section 3.2

In this part, we show the relationship between $Q_\alpha^*$, $\pi_\alpha^*$, and $Q_\alpha^{\pi_\alpha^*}$. Specifically, we show that under assumption 1 and 2

1. the optimal Q-function $Q_\alpha^* := \sup_{\pi \in \Pi} Q_\alpha^\pi$ is Lipschitz continuous on $\mathcal{A}$;

2. the optimal policy $\pi_\alpha^*$ exists, i.e., $\exists \pi_\alpha^* \in \Pi$ s.t. $Q_\alpha^{\pi_\alpha^*} = Q_\alpha^*$;

3. the relationship between $\pi_\alpha^*$ and $Q_\alpha^*$ is

$$\pi_\alpha^*(\cdot|s) = \arg\max_{q \in \Delta} \mathbb{E}_{a \sim q} [Q_\alpha^*(s, a)] + \alpha \mathcal{F}(q).$$

**Lemma 1.** *Under assumption 1, the set $\Delta$ is weakly compact.*

*Proof.* Denote

$$\mathcal{P}(\mathcal{A}) := \text{all Borel probability measures on } \mathcal{A},$$

where $\mathcal{A}$ is the compact action space. Then it comes from Theorem 9.4 in [50] that $\mathcal{P}(\mathcal{A})$ is weakly compact. By assumption 1, $\Delta$ is a weakly closed subset of $\mathcal{P}(\mathcal{A})$, so $\Delta$ is also weakly compact.    □

**Lemma 2.** *Under assumption 1, if the regularized Q-function $Q_\alpha(s, a)$ is continuous on the action space $\mathcal{A}$, then the optimization problem*

$$\max_{q \in \Delta} J_s(q) := \max_{q \in \Delta} \left( \mathbb{E}_{a \sim q}[Q_\alpha(s, a)] + \alpha \left( \mathbb{E}_{a \sim q}[f(a)] + \mathbb{E}_{a, a' \sim q}[g(a, a')] \right) \right) \qquad (11)$$

*always admits an optimal solution for all state $s \in \mathcal{S}$, where $f, g$ are defined in Equation (3).*

*Proof.* As $Q_\alpha, f, g$ are all bounded, we have

$$\max_{q \in \Delta} J_s(q) < +\infty.$$

Then we can find a sequence $(q_i)_{i=1}^\infty, q_i \in \Delta$ s.t.

$$\lim_{i \to +\infty} J_s(q_i) = \sup_{q \in \Delta} J_s(q).$$

By Lemma 1, $\Delta$ is weakly compact, so we can find a subsequence $(q_{i_j})_{j=1}^\infty$ that weakly converges to some $q^\infty \in \Delta$. Then by Theorem 2.8 in [5], we have

$$q_{i_j} \times q_{i_j} \Rightarrow q^\infty \times q^\infty.$$

As $Q_\alpha, f$ are both continuous on $\mathcal{A}$ and $g$ is continuous on $\mathcal{A} \times \mathcal{A}$, we obtain

$$J_s(q^\infty) = \lim_{j \to +\infty} J_s(q_{i_j}) = \sup_{q \in \Delta} J_s(q),$$

which means that $q^\infty$ is an optimal solution to $\max_{q \in \Delta} J_s(q)$.    □

**Lemma 3.** *Under Assumption 2, the optimal Q-function $Q_\alpha^* := \sup_{\pi \in \Pi} Q_\alpha^\pi$ is L-Lipschitz continuous on $\mathcal{A}$ for all $s \in \mathcal{S}$.*

*Proof.*

$$|Q_\alpha^*(s, a_1) - Q_\alpha^*(s, a_2)| = |\sup_{\pi \in \Pi} Q_\alpha^\pi(s, a_1) - \sup_{\pi \in \Pi} Q_\alpha^\pi(s, a_2)|$$
$$\leq \sup_{\pi \in \Pi} |Q_\alpha^\pi(s, a_1) - Q_\alpha^\pi(s, a_2)|$$
$$\leq L\|a_1 - a_2\|_2.$$

Thus, $Q_\alpha^*$ is Lipschitz continuous on $\mathcal{A}$ with Lipschitz constant $L$. $\qquad\square$

**Theorem 1.** *Under Assumption 1 and 2, the optimal policy $\pi_\alpha^*$ must exist and satisfy the following condition for all states:*

$$\pi_\alpha^*(\cdot|s) = \arg\max_{q \in \Delta} \mathbb{E}_{a \sim q} [Q_\alpha^*(s, a)] + \alpha \mathcal{F}(q). \tag{12}$$

*Proof.* By Lemma 3 we have that $Q_\alpha^*$ is Lipschitz continuous. Then by Lemma 2 there must exist some policy $\pi_\alpha^*$ s.t.

$$\pi_\alpha^*(\cdot|s) = \arg\max_{q \in \Delta} \left( \mathbb{E}_{a \sim q} [Q_\alpha^*(s, a)] + \alpha \left( \mathcal{F}(q) \right) \right), \ \forall\, s \in \mathcal{S}.$$

Then we have

$$Q_\alpha^*(s, a) = \max_{\pi \in \Pi} Q_\alpha^\pi(s, a)$$
$$= r(s, a) + \gamma \mathbb{E}_{s' \sim P(\cdot|s,a)} \left[ \max_{\pi \in \Pi} V_\alpha^\pi(s) \right]$$
$$= r(s, a) + \gamma \mathbb{E}_{s' \sim P(\cdot|s,a)} \left[ \max_{q \in \Delta} \left( \mathbb{E}_{a' \sim q} [\max_{\pi \in \Pi} Q_\alpha^\pi(s', a')] + \alpha \mathcal{F}(q) \right) \right]$$
$$= r(s, a) + \gamma \mathbb{E}_{s' \sim P(\cdot|s,a)} \left[ \max_{q \in \Delta} \left( \mathbb{E}_{a' \sim q} [Q_\alpha^*(s', a')] + \alpha \mathcal{F}(q) \right) \right]$$
$$= r(s, a) + \gamma \mathbb{E}_{s' \sim P(\cdot|s,a)} \left[ \mathbb{E}_{a' \sim \pi_\alpha^*(\cdot|s')} [Q_\alpha^*(s', a')] + \alpha \mathcal{F}(\pi^*(\cdot|s')) \right]$$
$$= \cdots$$
$$= Q_\alpha^{\pi_\alpha^*}(s, a), \ \forall\, (s, a) \in \mathcal{S} \times \mathcal{A}.$$

Hence $\pi_\alpha^*$ is the optimal policy of the regularized RL problem. $\qquad\square$

## A.2   Proof for Theorems in Section 3.3

In this part, we prove all the theorems in Section 3.3 for $d_\phi$.

**Theorem 2.** *Assume that any distribution $q \in \Delta$ is a product of its margin distributions. Then $\tilde{d}_\phi$ is a distance metric on the set of probability measures $\Delta$.*

*Proof.*

$$\tilde{d}_\phi^2(q, u) = 2\mathbb{E}_{a \sim q, a' \sim u}[\sum_{i=1}^{n} \phi\left(([a]_i - [a']_i)^2\right)]$$

$$- \mathbb{E}_{a,a' \sim u}[\sum_{i=1}^{n} \phi\left(([a]_i - [a']_i)^2\right)] - \mathbb{E}_{a,a' \sim q}[\sum_{i=1}^{n} \phi\left(([a]_i - [a']_i)^2\right)]$$

$$= \sum_{i=1}^{n} 2\mathbb{E}_{a \sim q, a' \sim u}[\phi\left(([a]_i - [a']_i)^2\right)]$$

$$- \sum_{i=1}^{n} \mathbb{E}_{a,a' \sim u}[\phi\left(([a]_i - [a']_i)^2\right)] - \sum_{i=1}^{n} \mathbb{E}_{a,a' \sim q}[\phi\left(([a]_i - [a']_i)^2\right)]$$

$$= \sum_{i=1}^{n} 2\mathbb{E}_{[a]_i \sim [q]_i, [a']_i \sim [u]_i}[\phi\left(([a]_i - [a']_i)^2\right)]$$

$$- \sum_{i=1}^{n} \mathbb{E}_{[a]_i, [a']_i \sim [u]_i}[\phi\left(([a]_i - [a']_i)^2\right)] - \sum_{i=1}^{n} \mathbb{E}_{[a]_i, [a']_i \sim [q]_i}[\phi\left(([a]_i - [a']_i)^2\right)]$$

$$= \sum_{i=1}^{n} d_\phi^2([q]_i, [u]_i),$$

where the third equation comes from that $q, u$ are both products of their own margin distributions. Previous work [3] has shown that $d_\phi([q]_i, [u]_i)$ is a metric on $[\Delta]_i$ for all $i = 1, 2, \cdots, n$. Thus, $\tilde{d}_\phi(q, u) = \left(\sum_{i=1}^{n} d_\phi^2([q]_i, [u]_i)\right)^{\frac{1}{2}}$ is also a metric on $\Delta$. $\qquad\square$

**Theorem 3.** *Assume the uniform distribution $u \in \Delta$. If $\mathcal{F}(q) = -d_\phi^2(q, u)$, then $\pi_\alpha^*(\cdot|s) \xrightarrow{d_\phi} u$ for all states when $\alpha \to \infty$.*

*Proof.* By Theorem 1 we have

$$V_\alpha^*(s) = \mathbb{E}_{a \sim \pi_\alpha^*(\cdot|s)}[Q_\alpha^*(s, a)] - \alpha d_\phi^2(\pi_\alpha^*(\cdot|s), u).$$

Define $\pi_u : \pi_u(\cdot|s) = u$. Then we have

$$0 \leq \frac{1}{\alpha}\left(V_\alpha^*(s) - V_\alpha^{\pi_u}(s)\right)$$

$$= \frac{1}{\alpha}\left(\mathbb{E}_{a \sim \pi_\alpha^*(\cdot|s)}[Q_\alpha^*(s, a)] - \mathbb{E}_{a \sim \pi_u(\cdot|s)}[Q_\alpha^{\pi_u}(s, a)]\right) - d_\phi^2(\pi_\alpha^*(\cdot|s), u)$$

$$\leq \frac{2\|Q_\alpha^*\|_\infty}{\alpha} - d_\phi^2(\pi_\alpha^*(\cdot|s), u).$$

As $Q_\alpha^*$ is bounded, when $\alpha \to \infty$, we have

$$d_\phi^2(\pi_\alpha^*(\cdot|s), u) \leq \frac{2\|Q_\alpha^*\|_\infty}{\alpha} \to 0,$$

which means $\pi_\alpha^*(\cdot|s) \xrightarrow{d_\phi} u$ for all states. $\qquad\square$

**Theorem 4.** *Suppose the action space $\mathcal{A}$ is convex and the set $\Delta$ contains all Borel probability measures on $\mathcal{A}$. Under Assumption 2, if we use the regularizer $\mathcal{F}(q) = -d_\phi^2(q, u)$ and choose function $\phi$ such that $\lim_{x \to 0^+} 2\alpha\sqrt{x}\phi'(x) > L$, then $\pi_\alpha^*(\cdot|s)$ will never be a Dirac delta distribution.*

*Proof.* Suppose that $|\mathcal{A}| > 1$. (Otherwise the problem is trivial.) Suppose $b \in \mathcal{A}$ is an arbitrary action. Choose another action $b_t \in \mathcal{A}$ such that $\|b - b_t\|_2 = t > 0$. Then $\delta_b(A) = \begin{cases} 0, & b \notin A \\ 1, & b \in A \end{cases}$ is a Dirac delta distribution on $\mathcal{A}$, and $q_t = \frac{1}{2}(\delta_b + \delta_{b_t})$ is a Dirac mixture distribution on $\mathcal{A}$.

To prove that $\pi_\alpha^*(\cdot|s)$ will never be a Dirac delta distribution for any state $s$, we only need to prove

$$\forall b \in \mathcal{A}, \ \exists b_t \in \mathcal{A} \ \text{ s.t. } \ J_s(q_t) > J_s(\delta_b),$$

where $J_s(q) = \mathbb{E}_{a\sim q}[Q_\alpha^*(s,a)] + \alpha\left(\mathbb{E}_{a\sim q}\left[f(a)\right] + \mathbb{E}_{a,a'\sim q}\left[g(a,a')\right]\right)$ is the regularized RL objective at state $s$ and $q_t$ is defined above.

Define

$$
\begin{aligned}
f(x) &:= -2\mathbb{E}_{y\sim u}[\phi(\|x-y\|_2^2)], \\
g(x,y) &:= \phi(\|x-y\|_2^2), \\
\Delta_Q(t) &:= \frac{1}{t}\left(\mathbb{E}_{a\sim q_t}[Q_\alpha^*(s,a)] - \mathbb{E}_{a\sim\delta_b}[Q_\alpha^*(s,a)]\right), \\
\Delta_f(t) &:= \frac{1}{t}\left(\mathbb{E}_{a\sim q_t}[f(a)] - \mathbb{E}_{a\sim\delta_b}[f(a)]\right), \\
\Delta_g(t) &:= \frac{1}{t}\left(\mathbb{E}_{a,a'\sim q_t}[g(a,a')] - \mathbb{E}_{a,a'\sim\delta_b}[g(a,a')]\right),
\end{aligned}
$$

then we have

$$J(q_t) - J(\delta_b) = t\left(\Delta_Q(t) + \alpha\left(\Delta_f(t) + \Delta_g(t)\right)\right).$$

For $\Delta_Q$: by Lemma 3 we have $Q_\alpha^*$ is Lipschitz, hence

$$
\begin{aligned}
\Delta_Q(t) &= \frac{1}{2t}\left(Q_\alpha^*(s,b_t) - Q_\alpha^*(s,b)\right) \\
&\geq -\frac{L}{2}.
\end{aligned}
$$

For $\Delta_f$: we have

$$\Delta_f(t) = \frac{1}{2t}\left(f(b_t) - f(b)\right).$$

Note that $f(\mathbf{x})$ is continuous differentiable on $\mathcal{A}$. We discuss the value of $\Delta_f(t)$ in three conditions.

1. If $\nabla f(b) = \mathbf{0}$, then we have $\Delta_f(t) = \frac{o(t)}{t}$, i.e., $\lim_{t\to 0^+} \Delta_f(t) = 0$.

2. If $\nabla f(b) \neq \mathbf{0}$ and $b \in \text{int}(\mathcal{A})$, then let $b_t = b + t\frac{\nabla f(b)}{\|\nabla f(b)\|_2}$, we have $\Delta_f(t) > 0$ for small enough $t$.

3. If $\nabla f(b) \neq \mathbf{0}$ and $b \in \text{bdry}(\mathcal{A})$, then we can find some $b_t$ s.t. $\nabla f^\top(b)(b_t - b) \geq 0$. To prove this, suppose $\nabla f^\top(b)(a-b) < 0$ for all $a \in \mathcal{A}$. Then define $b_\epsilon = b + \epsilon\nabla f(b)$ ($\epsilon > 0$), with small enough $\epsilon$, we have $f(b_\epsilon) \geq f(b)$. However, as $\|b_\epsilon - a\|_2 > \|b - a\|_2$ for all $a \in \mathcal{A}$, we have $f(b_\epsilon) < f(b)$ for all $\epsilon > 0$ by the definition of $f$, which results in a contradiction. Thus, we can find some $b_t$ s.t. $\nabla f^\top(b)(b_t - b) \geq 0$, and so we have $\underline{\lim}_{t\to 0^+} \Delta_f(t) \geq 0$.

Thus, $\forall \epsilon > 0$, we can always find some $b_t$ and $\delta_1$ s.t. $\forall 0 < t < \delta_1$ we have $\Delta_f(t) \geq -\epsilon$.

For $\Delta_g$: note that $\phi'$ is continuous and $\lim_{x\to 0^+} 2\alpha\sqrt{x}\phi'(x) > L$, then for small enough $t$, we have

$$
\begin{aligned}
\Delta_f(t) &= \frac{1}{2t}\left(g(b,b_t) - \phi(0)\right) \\
&= \frac{1}{2}\frac{\phi(t^2) - \phi(0)}{t} \\
&= \epsilon\phi'(\epsilon^2) \ (\epsilon \in (0,t)) \\
&> \frac{L}{2\alpha}.
\end{aligned}
$$

Therefore, we can always find some direction $(b_t - b)$ and some $\delta > 0$ s.t. for all $0 < t < \delta$, we have

$$J(q_t) - J(\delta_b) = t \left( \Delta_Q(t) + \alpha \left( \Delta_f(t) + \Delta_g(t) \right) \right)$$

$$> t \left( -\frac{L}{2} + \alpha \left( \Delta_f(t) - \epsilon \right) \right)$$

$$> t \left( -\frac{L}{2} + \alpha \left( \frac{L}{2\alpha} \right) \right)$$

$$= 0,$$

which means $\delta_b$ can never be the optimal policy at state $s$. Thus, the optimal policy $\pi_\alpha^*(\cdot|s)$ at any state $s \in \mathcal{S}$ will never be a Dirac distribution. $\square$

**Lemma 4.** *Assume the regularization function $\mathcal{F}$ maps $\Delta$ to a closed interval $[L_\mathcal{F}, U_\mathcal{F}]$. Then for any policy $\pi \in \Pi$, we have*

$$V^\pi + \alpha \frac{L_\mathcal{F}}{1 - \gamma} \leq V_\alpha^\pi \leq V^\pi + \alpha \frac{U_\mathcal{F}}{1 - \gamma}.$$

*Proof.* According to the definition of $V_\alpha^\pi(s)$, we have

$$V_\alpha^\pi(s) = \mathbb{E}_\pi \left[ \sum_{t=0}^\infty \gamma^t \left( r(s_t, a_t) + \alpha \mathcal{F} \left( \pi(\cdot|s_t) \right) \right) \middle| \mathbf{s_0} = s \right]$$

$$= \mathbb{E}_\pi \left[ \sum_{t=0}^\infty \gamma^t r(s_t, a_t) \middle| \mathbf{s_0} = s \right] + \alpha \mathbb{E}_\pi \left[ \sum_{t=0}^\infty \gamma^t \mathcal{F} \left( \pi(\cdot|s_t) \right) \middle| \mathbf{s_0} = s \right]$$

$$= V^\pi(s) + \alpha \mathbb{E}_\pi \left[ \sum_{t=0}^\infty \gamma^t \mathcal{F} \left( \pi(\cdot|s_t) \right) \middle| \mathbf{s_0} = s \right].$$

As $L_\mathcal{F} \leq \mathcal{F}(\pi(\cdot|s)) \leq U_\mathcal{F}$ for all state $s$, we have

$$\frac{1}{1 - \gamma} L_\mathcal{F} = \mathbb{E}_\pi \left[ \sum_{t=0}^\infty \gamma^t L_\mathcal{F} \middle| \mathbf{s_0} = s \right]$$

$$\leq \mathbb{E}_\pi \left[ \sum_{t=0}^\infty \gamma^t \mathcal{F} \left( \pi(\cdot|s_t) \right) \middle| \mathbf{s_0} = s \right]$$

$$\leq \mathbb{E}_\pi \left[ \sum_{t=0}^\infty \gamma^t U_\mathcal{F} \middle| \mathbf{s_0} = s \right]$$

$$= \frac{1}{1 - \gamma} U_\mathcal{F}.$$

Thus, we obtain

$$V^\pi(s) + \alpha \frac{L_\mathcal{F}}{1 - \gamma} \leq V_\alpha^\pi(s) \leq V^\pi(s) + \alpha \frac{U_\mathcal{F}}{1 - \gamma}, \ \forall s \in \mathcal{S}.$$

$\square$

**Theorem 5.** *Assume that the optimal policy $\pi_\alpha^*$ exists and the action space satisfies $\mathbf{diam}(\mathcal{A}) \leq U$. If we use the regularizer $\mathcal{F}(q) = -d_\phi^2(q, u)$, then we have*

$$V^* - \frac{2\alpha}{1 - \gamma} \left( \phi(U^2) - \phi(0) \right) \leq V^{\pi_\alpha^*} \leq V^*. \tag{13}$$

*Proof.* The right side of the inequality is obvious, as $\pi^*$ is the optimal policy of the original problem. For the left side, by Lemma 4 we have

$$V^* \leq V_\alpha^{\pi^*} - \alpha \frac{L_\mathcal{F}}{1 - \gamma}$$

$$V_\alpha^{\pi_\alpha^*} \leq V^{\pi_\alpha^*} + \alpha \frac{U_\mathcal{F}}{1 - \gamma}.$$

Hence we obtain

$$V^* \leq V_\alpha^{\pi^*} - \alpha \frac{L_\mathcal{F}}{1-\gamma}$$

$$\leq V_\alpha^{\pi_\alpha^*} - \alpha \frac{L_\mathcal{F}}{1-\gamma}$$

$$\leq V^{\pi_\alpha^*} - \alpha \frac{L_\mathcal{F}}{1-\gamma} + \alpha \frac{U_\mathcal{F}}{1-\gamma}.$$

The second inequality follows from that $\pi_\alpha^*$ is the optimal policy of the regularized problem. Define

$$g(x) = -\mathbb{E}_{a \sim u}[\phi(x,a)],$$

then $g(x)$ is continuous on $\mathcal{A}$, where $\mathcal{A}$ is a compact set. Let

$$a_0 = \arg\min_x g(x),$$

then we have

$$-d_\phi^2(\delta_{a_0}, u) \leq \mathcal{F}(q) \leq -d_\phi^2(u,u) = 0.$$

Note that

$$-d_\phi^2(\delta_{a_0}, u) = \mathbb{E}_{a,a' \sim \delta_{a_0}}[\phi(\|a-a'\|_2^2)] + \mathbb{E}_{a,a' \sim u}[\phi(\|a-a'\|_2^2)] - 2\mathbb{E}_{a \sim \delta_{a_0}, a' \sim u}[\phi(\|a-a'\|_2^2)]$$

$$= \phi(0) + \mathbb{E}_{a,a' \sim u}[\phi(\|a-a'\|_2^2)] - 2\mathbb{E}_{a \sim u}[\phi(\|a-a_0\|_2^2)]$$

$$\geq 2\left(\phi(0) - \phi(U^2)\right).$$

Thus, we obtain

$$V^* - \frac{2\alpha}{1-\gamma}\left(\phi(U^2) - \phi(0)\right) \leq V^{\pi_\alpha^*} \leq V^*.$$

$\square$

## A.3  Results for $\tilde{d}_\phi$

In this part, we provide similar results for $\tilde{d}_\phi$. The proofs for these results are similar to that for $d_\phi$. All results are under the assumptions

1. the action space $\mathcal{A}$ is a product space, i.e., $\mathcal{A} = \mathcal{A}_1 \times \mathcal{A}_2 \times \cdots \times \mathcal{A}_n$;

2. any distribution $q \in \Delta$ is a product of its margin distributions, i.e., $q = [q]_1 \times [q]_2 \times \cdots \times [q]_n$, where $[q]_i$ is the marginal distribution of $q$ on $\mathcal{A}_i$.

**Theorem 6.** *Assume the uniform distribution $u \in \Delta$. If $\mathcal{F}(q) = -\tilde{d}_\phi^2(q,u)$, then $\pi_\alpha^*(\cdot|s) \xrightarrow{\tilde{d}_\phi} u$ for all states when $\alpha \to \infty$.*

**Theorem 7.** *Suppose the action space $\mathcal{A}$ is convex and $\Delta = \{q = [q]_1 \times [q]_2 \times \cdots \times [q]_n \mid [q]_i \text{ is a Borel probability measure on } \mathcal{A}_i\}$. Under Assumption 2, if we use the regularizer $\mathcal{F}(q) = -\tilde{d}_\phi^2(q,u)$ and choose function $\phi$ such that $\lim_{x \to 0^+} 2\alpha\sqrt{x}\phi'(x) > L$, then the optimal policy distribution $\pi_\alpha^*(\cdot|s)$ for any state $s \in \mathcal{S}$ will never be a Dirac delta distribution.*

**Theorem 8.** *Assume that the optimal policy $\pi_\alpha^*$ exists and the action space satisfies $\mathbf{diam}(\mathcal{A}_i) \leq U$ for $i = 1, 2, \cdots, n$. If we use the regularizer $\mathcal{F}(q) = -\tilde{d}_\phi^2(q,u)$, then we have*

$$V^* - \frac{2n\alpha}{1-\gamma}\left(\phi(U^2) - \phi(0)\right) \leq V^{\pi_\alpha^*} \leq V^*. \tag{14}$$

## A.4  Policy Iteration for Regularized RL

In this section, we show the convergence of policy iteration for Regularized RL. All results in this section are under the Assumption 1 and 2.

**Lemma 5.** *The regularized Bellman operator and the regularized optimal Bellman operator are respectively defined by*

$$(\mathcal{T}_\alpha^\pi Q)(s,a) := r(s,a) + \gamma \mathbb{E}_{s' \sim P(\cdot|s,a)} \left[ (\mathcal{V}_\alpha^\pi Q)(s') \right], \tag{15}$$

$$(\mathcal{T}_\alpha^* Q)(s,a) := r(s,a) + \gamma \mathbb{E}_{s' \sim P(\cdot|s,a)} \left[ \sup_{\pi \in \Pi} (\mathcal{V}_\alpha^\pi Q)(s') \right], \tag{16}$$

$$\text{where } (\mathcal{V}_\alpha^\pi Q)(s') := \mathbb{E}_{a' \sim \pi(\cdot|s')} [Q(s',a')] + \alpha \mathcal{F}(\pi(\cdot|s')). \tag{17}$$

*Then both $\mathcal{T}_\alpha^\pi$ and $\mathcal{T}_\alpha^*$ are contraction operators.*

*Proof.* Let $Q_1, Q_2$ be two regularized Q-functions, then

$$|\mathcal{T}_\alpha^* Q_1(s,a) - \mathcal{T}_\alpha^* Q_2(s,a)|$$

$$= \left| r(s,a) + \gamma \left( \mathbb{E}_{s' \sim P(\cdot|s,a)} \left[ \sup_{\pi \in \Pi} (\mathcal{V}_\alpha^\pi Q_1)(s') \right] \right) - r(s,a) - \gamma \left( \mathbb{E}_{s' \sim P(\cdot|s,a)} \left[ \sup_{\pi \in \Pi} (\mathcal{V}_\alpha^\pi Q_2)(s') \right] \right) \right|$$

$$= \gamma \left| \mathbb{E}_{s' \sim P(\cdot|s,a)} \left[ \sup_{\pi \in \Pi} (\mathcal{V}_\alpha^\pi Q_1 - \mathcal{V}_\alpha^\pi Q_2)(s') \right] \right|$$

$$\leq \gamma \mathbb{E}_{s' \sim P(\cdot|s,a)} \left[ \sup_{\pi \in \Pi} |(\mathcal{V}_\alpha^\pi Q_1 - \mathcal{V}_\alpha^\pi Q_2)(s')| \right]$$

$$= \gamma \mathbb{E}_{s' \sim P(\cdot|s,a)} \left[ \sup_{q \in \Delta} (|\mathbb{E}_{a' \sim q} [Q_1(s',a') - Q_2(s',a')]|) \right]$$

$$\leq \gamma \mathbb{E}_{s' \sim P(\cdot|s,a)} \left[ \sup_{q \in \Delta} (\mathbb{E}_{a' \sim q} [|Q_1(s',a') - Q_2(s',a')|]) \right]$$

$$\leq \gamma \mathbb{E}_{s' \sim P(\cdot|s,a)} \left[ \sup_{q \in \Delta} (\mathbb{E}_{a' \sim q} [\|Q_1 - Q_2\|_\infty]) \right]$$

$$= \gamma \|Q_1 - Q_2\|_\infty,$$

and

$$|\mathcal{T}_\alpha^\pi Q_1(s,a) - \mathcal{T}_\alpha^\pi Q_2(s,a)|$$

$$= \left| r(s,a) + \gamma \left( \mathbb{E}_{s' \sim P(\cdot|s,a)} [(\mathcal{V}_\alpha^\pi Q_1)(s')] \right) - r(s,a) - \gamma \left( \mathbb{E}_{s' \sim P(\cdot|s,a)} [(\mathcal{V}_\alpha^\pi Q_2)(s')] \right) \right|$$

$$= \gamma \left| \mathbb{E}_{s' \sim P(\cdot|s,a)} [(\mathcal{V}_\alpha^\pi Q_1 - \mathcal{V}_\alpha^\pi Q_2)(s')] \right|$$

$$\leq \gamma \mathbb{E}_{s' \sim P(\cdot|s,a)} [|(\mathcal{V}_\alpha^\pi Q_1 - \mathcal{V}_\alpha^\pi Q_2)(s')|]$$

$$= \gamma \mathbb{E}_{s' \sim P(\cdot|s,a)} \left[ (|\mathbb{E}_{a' \sim \pi(\cdot|s')} [Q_1(s',a') - Q_2(s',a')]|) \right]$$

$$\leq \gamma \mathbb{E}_{s' \sim P(\cdot|s,a)} \left[ (\mathbb{E}_{a' \sim \pi(\cdot|s')} [|Q_1(s',a') - Q_2(s',a')|]) \right]$$

$$\leq \gamma \mathbb{E}_{s' \sim P(\cdot|s,a)} \left[ (\mathbb{E}_{a' \sim \pi(\cdot|s')} [\|Q_1 - Q_2\|_\infty]) \right]$$

$$= \gamma \|Q_1 - Q_2\|_\infty,$$

Thus, we have

$$\|\mathcal{T}_\alpha^* Q_1 - \mathcal{T}_\alpha^* Q_2\|_\infty \leq \gamma \|Q_1 - Q_2\|_\infty \text{ and } \|\mathcal{T}_\alpha^\pi Q_1 - \mathcal{T}_\alpha^\pi Q_2\|_\infty \leq \gamma \|Q_1 - Q_2\|_\infty,$$

which means both $\mathcal{T}_\alpha^\pi$ and $\mathcal{T}_\alpha^*$ are contraction mappings. $\square$

**Lemma 6.** *Suppose $\pi_{old} \in \Pi$. Let*

$$\pi_{new}(\cdot|s) = \underset{q \in \Delta}{\operatorname{argmax}} \, \mathbb{E}_{a \sim q} [Q_\alpha^{\pi_{old}}(s,a)] + \alpha \mathcal{F}(q), \ \forall s \in \mathcal{S}. \tag{18}$$

*Then we have $Q_\alpha^{\pi_{new}} \geq \mathcal{T}_\alpha^* Q_\alpha^{\pi_{old}} \geq Q_\alpha^{\pi_{old}}$.*

*Proof.* First we prove that $Q_\alpha^{\pi_{new}} \geq Q_\alpha^{\pi_{old}}$. As

$$\mathbb{E}_{a \sim \pi_{new}(\cdot|s)}[Q_\alpha^{\pi_{old}}(s,a)] + \alpha \mathcal{F}(\pi_{new}(\cdot|s)) \geq \mathbb{E}_{a \sim \pi_{old}(\cdot|s)}[Q_\alpha^{\pi_{old}}(s,a)] + \alpha \mathcal{F}(\pi_{old}(\cdot|s))$$
$$= V_\alpha^{\pi_{old}}(s),$$

we have

$$Q_\alpha^{\pi_{\text{old}}}(s,a) = r(s,a) + \gamma \mathbb{E}_{s' \sim P(\cdot|s,a)}[V_\alpha^{\pi_{\text{old}}}(s)]$$
$$\leq r(s,a) + \gamma \mathbb{E}_{s' \sim P(\cdot|s,a)}[\mathbb{E}_{a \sim \pi_{\text{new}}(\cdot|s)}[Q_\alpha^{\pi_{\text{old}}}(s,a)] + \mathcal{F}(\pi_{\text{new}}(\cdot|s))]$$
$$\leq \cdots$$
$$\leq Q_\alpha^{\pi_{\text{new}}}(s,a).$$

Thus, we have $Q_\alpha^{\pi_{\text{new}}} \geq Q_\alpha^{\pi_{\text{old}}}$. Now we start to prove that $Q_\alpha^{\pi_{\text{new}}} \geq \mathcal{T}_\alpha^* Q_\alpha^{\pi_{\text{old}}} \geq Q_\alpha^{\pi_{\text{old}}}$.

For the latter inequality, we have

$$\mathcal{T}_\alpha Q_\alpha^{\pi_{\text{old}}}(s,a) = r(s,a) + \gamma \left( \mathbb{E}_{s' \sim P(\cdot|s,a)}[\sup_{q \in \Delta} (\mathbb{E}_{a' \sim q}[Q_\alpha^{\pi_{\text{old}}}(s',a')] + \alpha \mathcal{F}(\pi_{\text{old}}(\cdot|s')))] \right)$$
$$\geq r(s,a) + \gamma \left( \mathbb{E}_{s' \sim P(\cdot|s,a)}[\mathbb{E}_{a' \sim \pi_{\text{old}}(\cdot|s')}[Q_\alpha^{\pi_{\text{old}}}(s',a')] + \alpha \mathcal{F}(\pi_{\text{old}}(\cdot|s'))] \right)$$
$$= Q_\alpha^{\pi_{\text{old}}}(s,a), \ \forall (s,a) \in \mathcal{S} \times \mathcal{A}.$$

For the former inequality, we have

$$Q_\alpha^{\pi_{\text{new}}}(s,a) = r(s,a) + \gamma \left( \mathbb{E}_{s' \sim P(\cdot|s,a)}[\mathbb{E}_{a' \sim \pi_{\text{new}}(\cdot|s')}[Q_\alpha^{\pi_{\text{new}}}(s',a')] + \alpha \mathcal{F}(\pi_{\text{new}}(\cdot|s'))] \right)$$
$$\geq r(s,a) + \gamma \left( \mathbb{E}_{s' \sim P(\cdot|s,a)}[\mathbb{E}_{a' \sim \pi_{\text{new}}(\cdot|s')}[Q_\alpha^{\pi_{\text{old}}}(s',a')] + \alpha \mathcal{F}(\pi_{\text{new}}(\cdot|s'))] \right)$$
$$= r(s,a) + \gamma \left( \mathbb{E}_{s' \sim P(\cdot|s,a)}[\sup_{q \in \Delta} (\mathbb{E}_{a' \sim q}[Q_\alpha^{\pi_{\text{old}}}(s',a')] + \alpha \mathcal{F}(q))] \right)$$
$$= \mathcal{T}_\alpha Q_\alpha^{\pi_{\text{old}}}(s,a), \ \forall (s,a) \in \mathcal{S} \times \mathcal{A}.$$

Thus, we have $Q_\alpha^{\pi_{\text{new}}} \geq \mathcal{T}_\alpha^* Q_\alpha^{\pi_{\text{old}}} \geq Q_\alpha^{\pi_{\text{old}}}$. $\qquad \square$

**Lemma 7.** $Q_\alpha$ *is the optimal Q-function of the regularized RL if and only if $Q_\alpha$ satisfies the Bellman optimality condition. That is,*

$$Q_\alpha = \sup_{\pi \in \Pi} Q_\alpha^\pi \Leftrightarrow Q_\alpha = \mathcal{T}_\alpha^* Q_\alpha. \qquad (19)$$

*Proof.* Suppose $Q_\alpha = \sup_{\pi \in \Pi} Q_\alpha^\pi$. By Lemma 6, we have

$$\mathcal{T}_\alpha^* Q_\alpha(s,a) \geq Q_\alpha(s,a).$$

By Theorem 1, there must exist some policy $\pi_\alpha^* \in \Pi$ s.t. $Q_\alpha^{\pi_\alpha^*} = Q_\alpha$. Then

$$\mathcal{T}_\alpha^* Q_\alpha(s,a) = \mathcal{T}_\alpha^* Q_\alpha^{\pi_\alpha^*}(s,a) \leq Q_\alpha^{\pi_{\text{new}}^*}(s,a) \leq \sup_{\pi \in \Pi} Q_\alpha^\pi(s,a) = Q_\alpha(s,a),$$

where the first inequality comes from Lemma 6. Thus, we have $\mathcal{T}_\alpha^* Q_\alpha(s,a) = Q_\alpha(s,a)$.

Note that $\mathcal{T}_\alpha^*$ is a contraction mapping, so the solution to $Q = \mathcal{T}_\alpha^* Q$ is unique. Thus, we have

$$Q_\alpha = \sup_{\pi \in \Pi} Q_\alpha^\pi \Leftrightarrow Q_\alpha = \mathcal{T}_\alpha^* Q_\alpha.$$

$\qquad \square$

**Lemma 8** (Policy Evaluation). *Let $Q_\alpha^0 : \mathcal{S} \times \mathcal{A} \to \mathbb{R}$ be an arbitrary bounded function. For any policy $\pi \in \Pi$, define $Q_\alpha^{k+1} := \mathcal{T}_\alpha^\pi Q_\alpha^k$. Then we have $\|Q_\alpha^\pi - Q_\alpha^k\|_\infty \leq \gamma^k \|Q_\alpha^\pi - Q_\alpha^0\|_\infty$.*

*Proof.* According to the definition of $Q_\alpha^\pi$, we have

$$\mathcal{T}_\alpha^\pi Q_\alpha^\pi = Q_\alpha^\pi.$$

That is, $Q_\alpha^\pi$ is the fixed point of $\mathcal{T}_\alpha^\pi$. Hence

$$\|Q_\alpha^\pi - Q_\alpha^k\|_\infty = \|\mathcal{T}_\alpha^\pi Q_\alpha^\pi - \mathcal{T}_\alpha^\pi Q_\alpha^{k-1}\|_\infty$$
$$\leq \gamma \|Q_\alpha^\pi - Q_\alpha^{k-1}\|_\infty$$
$$\leq \cdots$$
$$\leq \gamma^k \|Q_\alpha^\pi - Q_\alpha^0\|_\infty.$$

$\qquad \square$

**Theorem 9** (Policy Iteration). *Suppose $\pi_0 \in \Pi$, and a sequence of policies $(\pi_k)_{k=0}^{\infty}$ is obtained by repeatedly updating the policy via Equation (18). Then we have $\|Q_\alpha^* - Q_\alpha^{\pi_k}\|_\infty \leq \gamma^k \|Q_\alpha^* - Q_\alpha^{\pi_0}\|_\infty$.*

*Proof.* By Lemma 6 and Lemma 7, we have

$$
\begin{aligned}
\|Q_\alpha^* - Q_\alpha^{\pi_k}\|_\infty &\leq \|Q_\alpha^* - \mathcal{T}_\alpha^* Q_\alpha^{\pi_{k-1}}\|_\infty \\
&= \|\mathcal{T}_\alpha^* Q_\alpha^* - \mathcal{T}_\alpha^* Q_\alpha^{\pi_{k-1}}\|_\infty \\
&\leq \gamma \|Q_\alpha^* - Q_\alpha^{\pi_{k-1}}\|_\infty \\
&\leq \cdots \\
&\leq \gamma^k \|Q_\alpha^* - Q_\alpha^{\pi_0}\|_\infty.
\end{aligned}
$$

$\square$

# B  Details of Algorithm Implementation and Experiment Settings

## B.1  Regularization Term

Here we provide details of our regularizers used in this paper, including Gini mean difference (GMD), generalized energy distance (GED) , and internal energy (IE).

Table 5: Expressions of $f, g$ for our regularizers. In this table, $[\delta]_i$ denotes $|[a]_i - [a']_i|$, $[b]_i$ denotes $1 + [a]_i$ and $[c]_i$ denotes $1 - [a]_i$. In practice, we impose a small $\epsilon > 0$ on $[\delta]_i$, $[b]_i$ and $[c]_i$

| $\mathcal{F}(q)$ | $\phi(z)$ | $f(a)$ | $g(a, a')$ |
|---|---|---|---|
| GMD | $z^{0.5}$ | - | $\sum_{i=1}^{n} [\delta]_i$ |
| GED | $z^{0.5}$ | $-2\sum_{i=1}^{n}\left([a]_i^2 + 1\right)$ | $\sum_{i=1}^{n} [\delta]_i$ |
|  | $z^{0.25}$ | $-\frac{4}{3}\sum_{i=1}^{n}\left([b]_i^{1.5} + [c]_i^{1.5}\right)$ | $\sum_{i=1}^{n} \sqrt{[\delta]_i}$ |
|  | $\log z$ | $-2\sum_{i=1}^{n}\left([c]_i \log[c]_i - [b]_i \log[b]_i\right)$ | $\sum_{i=1}^{n} \log[\delta]_i^2$ |
| IE | $z^{0.25}$ | - | $\sum_{i=1}^{n} \sqrt{[\delta]_i}$ |

## B.2  Policy Networks

Here, we introduce the policy architectures used in the Section 5. For each policy architecture, we apply the function $\mathrm{tanh}$ to the outputs since the action spaces are bounded in $[-1, 1]^n$.

**Squashed Gaussian (SG)**   We use the same policy architecture as that used in SAC [20]. The actions are sampled from a Gaussian whose means and covariances are given by a neural network.

**Noisy network (NN)**   The policy is represented by a neural network [11] with parametric noise added to its weights. In our experiments, we use the factorised Gaussian noise. We refer the interested readers to the paper [11] for more details.

**Dirac mixtures (DM)**   Each dimension of the action is randomly selected from 32 possible values with equal probability. Therefore, for each state, the network outputs a vector with $32 \times n$ dimensions. Moreover, each dimension of action is independent of other dimensions.

**Generator Model (GM)**   The policy is represented by a neural network with noise added to its last layer. Moreover, all dimensions of the actions are also sampled independently.

**Normalizing flow (NF)**   We use the same policy architecture as that used in SAC-NF [28]. We use the *radial* [38] flows since it achieves better performance than *planar* [38] and *IAF* flows.

## B.3  Details of Experiment Settings and Hyperparameters

**Reproducing the Baselines**   We list the implementations of our baselines, including soft actor-critic (SAC), twin delayed deep deterministic policy gradient algorithm (TD3), and deep deterministic policy gradient (DDPG).

**SAC** We use the pyTorch implementation of SAC in `https://github.com/vitchyr/rlkit`, which is recommended by the authors of SAC.

**TD3** We use the up-to-date code from the authors' git repository `https://github.com/sfujim/TD3`. The authors have tuned the hyperparameters several times after releasing the code and thus the current version is much stronger than the one used in their paper [12].

**DDPG** We use the pyTorch implementation of DDPG in `https://github.com/vitchyr/rlkit`.

**Hyperparameters** We use the same hyperparameter as that of SAC [20] if possible. Table 6 lists the common ACED parameters used in the comparative evaluation and abalation study.

Table 6: Shared parameters used in all experiments.Here, $\mathcal{N}(\mathbf{0}, 0.09\mathbf{I})$ denotes a Gaussian distribution with mean $\mathbf{0}$ and covariance matrix $0.09\mathbf{I}$.

| Parameter | Value |
|---|---|
| optimizer | Adam [23] |
| learning rate | $3 \cdot 10^{-4}$ |
| discount ($\gamma$) | 0.99 |
| replay buffer size | $10^6$ |
| number of hidden layers | 2 (1 for NF policies) |
| number of hidden units per layer | 256 |
| number of samples per minibatch | 256 |
| nonlinearity | ReLU |
| target smoothing coefficient | 0.005 |
| target update interval | 1 |
| sample number | 32 |
| target valur (T) | $\mathcal{F}(\mathcal{N}(\mathbf{0}, 0.09\mathbf{I}))$ |

**Evaluation policies** As with SAC, we use a deterministic policy for evaluation. Except DM policy, all policy can be regard as $h(s, \xi)$ where $\xi$ is sampled from a standard normal distribution. For these networks, the corresponding evaluation policy is represented by $h(s, \mathbf{0})$. For DM policy, the deterministic policy outputs the mean of actions. We provide the performance comparison between the stochastic policy and evaluation policy in the next section.

## C  Additional Experimental Results

### C.1  Evaluation policy

We provide comparison between the stochastic policies and the evaluation policies. The results in Figure 4 show that using evaluation policies results in a higher return.

### C.2  Discussion on performance

ACED (**power-0.25**) outperforms SAC even with SG policies (same as that used in SAC). To understand how the performance difference comes, we analyze the stochasticity of learned policies. For each policy, we sample 20000 states and estimate the entropies of policy distributions at these states. Figure 5 is the histogram of these entropies. This histogram shows that our regularization **power-0.25** leads to higher variability of entropies compared to SAC.

### C.3  Performance Comparison for Normalizing FLow Policies

We compare ACED and SAC with normalizing flow policies. We use authors' implementation of SAC-NF [28] in this Section. Results in Figure 6 show that ACED achieves comparable performance with SAC. Moreover, ACED is more efficient that SAC (shown in Table 4) when using NF policies.

Figure 4: Comparison between the stochastic policies of ACED and the evaluation policies.

Figure 5: Comparison of stochasticity between ACED and SAC. We use this histogram to show distribution of entropies.

## C.4 Examples in Multi-Goal Environments

We use the similar example as in soft Q-learning [17] to show that ACED can learn multi-modal distributions. Figure 7 illustrates the 2D multi-goal environments and the learned policy. The left shows the reward functions and the trajectories from a policy learned with ACED. The right shows the Q-values and the sampled actions at state $(0, 0)$.

Figure 6: Comparison of performance between ACED and SAC (with normalizing flow policies). We run each experiments with three different random seeds.

(a) Sample number

(b) Policy architecture

Figure 7: Learned policy with ACED. The left illustrates the trajectories from the learned policy. The right shows the sampled actions at state $(0, 0)$. Results show that ACED can learn a policy that captures multiple goals. Moreover, the right picture shows the our regularization can lead to multimodal behaviors.

## C.5 Comparison Between $d_\phi$ and $\tilde{d}_\phi$

We compare the regularization defined by $d_\phi$ with the regularization defined by $\tilde{d}_\phi$ (see Figure 8). We approximate the function $f$ in the regularization $d_\phi^2$ by sampling from the uniform distribution $u$. The reason why the regularization $d_\phi^2$ does not work is that sampling from the uniform distribution introduces large variance of the estimated gradient.

Figure 8: Comparison between the regularization defined by $d_\phi$ with the regularization defined by $\tilde{d}_\phi$.

## C.6 Sensitivity Analysis

ACED is insensitive to the sample number $N$ and the target value $T$. The sensitivity analysis for $N$ can be found in Section 5.3 and Appendix C.9. Figure 9 shows the sensitivity analysis for $T$. Here, we set $T$ as $\mathcal{F}(\mathcal{N}(\mathbf{0}, \lambda \mathbf{I}))$, where $\mathcal{N}(\mathbf{0}, \lambda \mathbf{I})$ is a Gaussian distribution with the covariance matrix $\lambda \mathbf{I}$.

Figure 9: Sensitivity analysis for target value $T$.

Table 7: Comparison between ACED and SAC. Both algorithms use ensemble Gaussian policy whose ensemble size is 5. We record the best performance during the first 1.0 million steps.

| Algorithm | ACED ($N = 2$) | SAC |
|---|---|---|
| **Time (h)** | $7.06 \pm 0.30$ | $10.96 \pm 1.34$ |
| **Performance** | $4697 \pm 1140$ | $3641 \pm 917$ |

### C.7 Comparison with Generative Actor Critic (GAC)

We compare ACED with GAC [48] in HalfCheetah-v2 task. GAC also does not limit the representation of the policy. Figure 10 shows that our method outperforms GAC. We note that GAC is expensive in computation. GAC takes more than $65h$ to train the policy with about 1.0 million steps, while ACED takes less than $8h$.

Figure 10: Comparison of performance between ACED and GAC in HalfCheetah-v2.

### C.8 Comparison between ACED and SAC in Efficiency

To show the improved efficiency of ACED, we evaluate ACED ($N = 2$) against SAC with an ensemble of policies in HalfCHeetah-v2 task. The ensemble size is 5, and each policy outputs a Gaussian distribution. Experiments 7 show runing ACED with 1 million samples costs about 7 hours while runing SAC costs about 11 hours. The core reason is that ACED does not need to compute probability density, which requires the forward prediction of all networks.

### C.9 More Results for Ablation Study

We provide additional results (Figure 11, 12 and 13) as a supplement to Section 5.3. The results in Ant-v2 tasks further support our conclusions in Section 5.3. In Humanoid-v2 task, Using regularization does not lead to significant performance improvements.

Figure 11: Performance of ACED when sample number $N$ varies.

Figure 12: Performance of ACED with different policy architectures.

Figure 13: Performance of ACED with different regularizers.