[Reviews · NeurIPS 2020]

Review 1

Summary and Contributions: The authors propose a novel regularization term for policies in reinforcement learning. This regularization can be easily computed based on samples. This is in particular important for expressive policies with non-Gaussian action distributions, where entropy based regularizeres, or in general regularizeres that require the evaluation of the likelihood, would require much more involved inference procedures. The proposed regularization term is employed in an Actor-Critic DRL framework and compared to state-of-the-art RL methods. Theoretical results support the validity of the proposed policy regularization. Empirical results depict the state-of-the-art performance even with complex policy classes with improved computational efficiency

Strengths: The proposed sample based regularization term seems to be a rather heuristic but still interesting method to achieve policy regularization with low computational overhead. The authors discuss the applicability of the proposed regularization term for different (more or less expressive) classes of policies and empirically evaluate their learning performance with and without regularization The authors demonstrate interesting examples of GED based regularizers to promote stochasticity, enable multimodal behaviors and incorporate geometric information. It would be interesting to visualize the failure modes of entropy regularized or non-regularized methods in comparison to fully appreciate the benefits of the GED based regularizers.

Weaknesses: The proposed ACED algorithm seems to resemble SAC exactly, except for replacing the entropy based regularization with the proposed sample based regularizer (SBR). From the current presentation, it is unclear to me whether or not there are other subtle differences between SAC and ACED that might cause performance differences. The presentation of the actual AC method seems peripheral to the discussion about the actual regularization method. The SBR is presented and evaluated specifically in the context of the proposed actor critic (ACED) method. From the general presentation, I would expect the regularizer to be applicable in a much wider scope of RL methods. I would be interested in learning about the general applicability or limitations of the proposed regularizer and possibly evaluations of its efficiency when employed in other RL methods. The presented result demonstrate state-of-the-art performance and demonstrate the performance increase when utilizing regularization. However, there is only a marginal performance improvement to state-of-the-art for some of the proposed, heuristic regularization schemes (log, power-0.25). The main benefit is visible for more complex policies (normalizing flow) in terms of computational efficiency but not so visible for the rest. Theorem 5 proposes bound on the regularized value function. Whilst being an interesting contribution, I am missing a discussion about its implication and interpretation.

Correctness: The presented theoretical results are properly derived in the appendix. The empirical evaluation supports the main claims and additional information is available in the appendix.

Clarity: The paper is well written and easy to follow. Notation is consistent.

Relation to Prior Work: The relation and differences to entropy based regularization is clearly presented. A short overview of RL regularization is given, which however mainly focuses on entropy style regularization. Related work on applicable policy classes is discussed.

Reproducibility: Yes

Additional Feedback: From the current presentation, it was not immediately obviously to me, why the two SBR instances (table 2) are presented. i.e. when would we have a distribution as a product of its marginal? Do we see a comparison/evaluation? When do you employ the one/the other? The presentation of the main idea feels a bit scattered between section 2.3 (Generalized Energy Distance), section 3.2 (sample-based regularization), table 1 and table 2 and interleaved with theoretical insights and the differentiation to entropy-based regularizers. Maybe this presentation could be restructured to help the presentation of the main contribution. Section 3.3 is called Examples of Sample-Based Regularization but presents theorems about the SBR validity/behavior.


Review 2

Summary and Contributions: The paper proposed a sample-based regularization solution for regularized reinforcement learning. Compared with entropy-based regularization such as SAC, the proposed solution better promotes policy stochasticity. It also supports a richer and more complex set of policy representations and faster training speed with comparable policy performance. The sample-based regularization is based on the generalized energy distance (GED) between the learned policy and a policy that samples action uniformly.

Strengths: POST REBUTTAL: I agree with other reviewers that the improvements in the experiments are relatively small. Meanwhile, the authors didn't demonstrate a case in which gaussian policy fails but ACED succeeds. However, the authors showed the benefits in terms of run time in the paper and the rebuttal. Moreover, I think methodological this paper has its own novelty and contribution to the RL community. Therefore, I vote for acceptance. The paper has clear contribution to the RL community. The proposed sample-based regularization is novel. It enables the possibilities to use a wider range of policy representations. Meanwhile, it is computational more efficient; ablation study (Figure 3 (a)) shows that two samples are enough to estimate the regularization term and achieve good performance. Also, the paper is well written.

Weaknesses: One advantage of this method is that it supports more complex policy representations. The authors should try to find more appealing RL tasks in which such advantage leads to better performance. In Table 2, DM does not show better performance than SG. The paper proposed two forms of regularizer: \tilde{d}^2_\phi and d^2_\phi. While the former gives a closed-form equation of function $f$, the latter needs to approximate $f$ by sampling. However, in terms of task performance in the experiments, which one is better? Figure 5 in appendix shows the histogram of entropies of ACED and SAC. It would be more interesting to plot this histogram every N steps (e.g. 10^5 steps) during training, just to confirm that the difference of stochasticity between the two algorithms is consistent, especially in the early stage of training.

Correctness: yes

Clarity: yes

Relation to Prior Work: yes

Reproducibility: Yes

Additional Feedback:


Review 3

Summary and Contributions: The paper proposes a new way to regularize policies in continuous action space to make them stochastic and thus to enhance exploration. Unlike entropy-based regularization, the proposed regularization is sample based and does not need to estimate the density function of the policy. This feature allows the proposed method to be applicable to more complex forms of policies. The paper demonstrates some theoretical features of the proposed regularization, and outlines an agent equipped with this form of regularization, which is then evaluated empirically on a range of environments.

Strengths: - The new form of regularization seems interesting, and is application to complex policy classes in continuous action space. The work could be interesting to the reinforcement learning community. - The experiments seem pretty thorough.

Weaknesses: - From the experimental results, it seems that the improvement is not very significant. I think a main feature of this sample based regularization is that you could use a more complex policy representation that would be computationally infeasible for other entropy based methods. It would be great to have an example of that. - I think as a weakness of policy gradient methods with regularization in general, the exploration is local and shortsighted. The agent cannot demonstrate behaviors such as making a concerted effort to explore a certain part of the state space that it is uncertain about. For some problems, it is conceivable that policy gradient with a well-regularized stochastic policy may be sufficient for exploration, but I think that in general, such exploration methods are quite inefficient, and it is easy to cook up toy examples where such methods wound require exponentially amount of time to explore.

Correctness: In line 204, the authors say they did not tune the hyperparameters. I am not sure if they are referring to their algorithm or the baselines. If it is the latter, I would cast doubt on the experimental results.

Clarity: The paper is clear and well-organized.

Relation to Prior Work: Yes.

Reproducibility: Yes

Additional Feedback:


Review 4

Summary and Contributions: Thank you for answering many of my questions! Despite promising results and motivation, some of my concerns have not been addressed, i.e., providing tasks where a Gaussian policy failed to learn with SAC while ACED can solve. After reading the authors' feedback, I choose to maintain my score. ----- The limitation of entropy regularization in successful RL algorithms such as SAC is that it requires the function density of the policy. This authors propose a new regularization term that applies to other forms of policy without its density. The key idea is to encourage sampled actions to stay away from each other; therefore, achieving the exploration incentive.

Strengths: The experiment sections are done quite thoroughly. I like the study of comparison between different policy architectures and the phi function. Theorem 5 seems to offer the notion of the cost of exploration for optimality under some assumptions. The authors however do not make claims or discuss further --making it difficult to understand its significance.

Weaknesses: Despite interesting study on standard benchmark in this paper, I think it is missing environments where we can clearly see that SAC fails and ACED succeeds perhaps in multi-modal settings. The authors may also compare against policy gradient methods which do not necessary require specific form of policy. Should the community switch from SAC to ACED? To answer this, perhaps we can ask how robust/sensitive the algorithm is to hyperparameters. Gaining this insight will be valuable.

Correctness: I am convinced by the experimental results and the proofs.

Clarity: The paper is a little bit hard to follow. The authors should explain the motivation behind each subsection. The paper requires another pass to remove typos such as "The" in the description of figure 1. It seems like theorem 5 is a major contribution of this paper but there is no explanation or claims around it.

Relation to Prior Work: Related work does not draw connections to the work the author proposes, and therefore it is difficult to see the novelty contribution of this approach.

Reproducibility: Yes

Additional Feedback: What is Gini mean difference? Please give explanation. What is the different between figure 1(a) and 1(d)? What does the notation of phi^(n+1) mean on line 66?

[Author Response · NeurIPS 2020]



Figure A

Figure B

Figure C

Figure D

We thank the reviewers for insightful comments. We are improving the paper by incorporating the reviewers' suggestions.

**Response to reviewer #1:**

**Why we present two SBR instances (Table 2)** The first $d_\phi^2$ requires sampling from a uniform distribution to approx-
imate $f$, which leads to a large variance of gradient and then hurts the performance (see Figure A). Therefore, we
propose $\tilde{d}_\phi^2$ to reduce the variance, which provides a closed-form expression of $f$ and does not require sampling. We
will add more discussions about these two instances in the final version, if accepted.

**Discussion about Theorem 5** Theorem 5 proposes a bound on the performance gap between the regularized optimal
policy $\pi_\alpha^*$ and the original optimal policy $\pi^*$. This theorem shows that the gap depends on $\Delta = \phi(U^2) - \phi(0)$. Though
larger $\Delta$ can encourage actions away from each other better, it may lead to worse performance of $\pi_\alpha^*$. We will add the
discussion about Theorem 5 in the final version, if accepted.

**Performance difference between SAC and our method (ACED)** When we use Gaussian policies, the hyperparam-
eters of ACED (Table 6 in Appendix) are the same as that of SAC except for the target value $T$. Therefore, the
performance difference mainly comes from different regularization.

**More examples that demonstrate the benefits of ACED** To show the improved efficiency of ACED, we evaluate
ACED ($N = 2$) against SAC with an ensemble of policies. The ensemble size is 5, and each policy outputs a Gaussian
distribution. Experiments show that 1000 updates in ACED cost 21.4s while those in SAC cost 33.7s. The core reason
is that ACED does not need to compute probability density, which requires the forward prediction of all networks. We
will provide results in details in the final version, if accepted.

**Response to reviewer #2:**

**Compare the two SBR instances** In practice, $\tilde{d}_\phi^2$ is better than $d_\phi^2$. Please see the response to reviewer #1 for details.

**Histograms of entropies of ACED and SAC** In Figure B, we provide the histograms of entropies of SAC and ACED
during training. It shows that the difference of stochasticity exists through almost the whole training procedure.

**Response to reviewer #3:**

**The meaning of "we did not tune the hyperparameters"** For SAC and TD3, we use the hyperparameters provided
by their authors. Therefore, we did not tune them again. For our algorithm, we use the same hyperparameters (see
Table 6 in Appendix) as SAC, if possible. We did not tune $N$ and $T$ for the results in Section 5.1.

**Examples of complex policies** We provided evaluation with normalizing flow policies as an example in Section 5.2.
Moreover, expressing the policies by a noisy network [10] requires an additional classification network to estimate
entropy [44]. We will provide more examples in the final version, if accepted.

**Response to reviewer #4:**

**Discussion about Theorem 5** Please refer to the response to reviewer #1.

**Should the community switch from SAC to ACED?** If computing the probability density of policies is time-
consuming or infeasible, the answer is "yes". For example, when parameterizing policies by noisy networks, ACED is a
better choice than SAC. Otherwise, the answer depends on the performance of SAC and ACED.

**How sensitive ACED is to hyperparameters?** ACED is insensitive to the sample number $N$ and the target value $T$.
The sensitivity analysis for $N$ can be found in Section 5.3 and Appendix C.5. Figure C shows the sensitivity analysis
for $T$. Here, we set $T$ as $\mathcal{F}(\mathcal{N}(\mathbf{0}, \lambda\mathbf{I}))$, where $\mathcal{N}(\mathbf{0}, \lambda\mathbf{I})$ is a Gaussian distribution with the covariance matrix $\lambda\mathbf{I}$.

**Connections with related work and our novelty contributions** Most existing regularization [18,23,51] takes the
form of $\mathbb{E}_{a\sim\pi(\cdot|s)}[f(\pi(a|s))]$. We propose a novel regularization form (Equation 4) to encourage stochasticity. Due to
the different forms, previous regularization often requires computing probability density to estimate entropy but our
regularization does not. Moreover, unlike previous regularization, our regularization considers the distances between
actions and thus incorporates geometric information.

**What is Gini mean difference?** Gini mean difference is a measure of statistical dispersion. Given a distribution $D$, it
is defined by $\mathbb{E}|X_1 - X_2|$, where $X_1$ and $X_2$ are sampled from $D$ independently.

**What is the difference between Figures 1(a) and 1(d)?** The reward function in Figure 1(a) is unimodal, while the
other is trimodal. Figure 1(d) shows that our regularization can lead to a multi-modal distribution.

**What does the notation of $\phi^{(n+1)}$ mean on line 66?** $\phi^{(n+1)}$ denote the $(n + 1)$th order derivative of $\phi$.

**Comparison with methods that do not require a specific form of policy:** We compare ACED with GAC [47] in
HalfCheetah-v2 . Figure D shows that our method outperforms GAC. We note that GAC is expensive in computation.
GAC takes more than 55h to train the policy with about 0.8 million steps, while ACED takes less about 6h. As the
experiments are still running, we will provide more results in the final version, if accepted.

[Meta-Review · NeurIPS 2020]

Seems like this is a meaningful contribution of the RL community, although the accept was not unanimously. Please correct the manuscript according to the reviewers requests. I thank both the authors and reviewers for their efforts!